# Layer-by-Layer Self-Assembly Coating for Multi-Functionalized Fabrics: A Scientometric Analysis in CiteSpace (2005–2021)

**DOI:** 10.3390/molecules27196767

**Published:** 2022-10-10

**Authors:** Ying Pan, Li Fu, Jia Du, Dong Zhang, Ting Lü, Yan Zhang, Hongting Zhao

**Affiliations:** 1Institute of Environmental Materials and Applications, College of Materials and Environmental Engineering, Hangzhou Dianzi University, Hangzhou 310018, China; 2Key Laboratory of Advanced Textile Materials and Manufacturing Technology and Engineering Research Center for Eco-Dyeing & Finishing of Textiles, Zhejiang Sci-Tech University, Hangzhou 310018, China; 3Key Laboratory of Green Cleaning Technology & Detergent of Zhejiang Province, Lishui 323000, China; 4School of Environmental and Chemical Engineering, Foshan University, Foshan 528011, China

**Keywords:** layer-by-layer self-assembly, coating, fabric, functionalization, bibliometrics

## Abstract

Surface-engineered coatings have been increasingly applied to functionalize fabrics due to the ease of deposition of the coatings and their effectiveness in endowing the fabric with abundant properties. Among the surface modification methods, layer-by-layer (LbL) self-assembly has emerged as an important approach for creating multifunctional surfaces on fabrics. In this review, bibliometric analysis with the visualization analysis of LbL self-assembly coatings on fabrics was performed on publications extracted from the Web of Science (WOS) from 2005 to 2021 based on the CiteSpace software. The analysis results showed that research on LbL self-assembly coatings on fabrics has attracted much attention, and this technique has plentiful and flexible applications. Moreover, research on the LbL self-assembly method in the field of functionalization of fabrics has been summarized, which include flame retardant fabric, antibacterial fabric, ultraviolet resistant fabric, hydrophobic fabric and electromagnetic shielding fabric. It was found that the functionalization of the fabric has been changing from singularity to diversification. Based on the review, several future research directions can be proposed. The weatherability, comfort, cost and environmental friendliness should be considered when the multifunctional coatings are designed.

## 1. Introduction

As early as 1966, Iler proposed the layer-by-layer (LbL) self-assembly technology by alternately depositing two oppositely charged species [1]. After a long time, this technology was utilized to prepare multilayer films composed of polyelectrolytes on solid substrates in 1992 [2]. Since then, researchers have applied the LbL self-assembly method to conductive membranes [3,4,5], permeable membranes [6,7] and surface modification [8,9,10]. Moreover, the coordination [11], hydrogen bond [12] and specific molecular recognition [13] have been used as driving forces during the LbL process. The components in the multilayer films extended from polyelectrolyte to nanoparticles, micelles, etc. [14]. Due to the excellent performance of the LbL self-assembly method in the preparation process and its obtained properties, it has gradually become an important technology for modifying the surfaces of polymer matrixes, which include polyurethane foam [15], cotton [16], poly (ethylene terephathalate) fabric and film [17], polystyrene microsphere [18], etc. In recent years, LbL self-assembly technology has been widely used for fabricating the coating with the properties of gas barrier [19], conductivity [18], hydrophobicity [20] and flame retardancy [21].

The fabric shows a large specific surface area. Lots of technologies have been employed to impart multifunctional properties to textiles. These technologies include pad-dry-cure process, sol-gel method, UV light grafting method, atom transfer radical polymerization, etc. [22,23,24,25,26,27,28,29,30]. Compared with traditional finishing methods, LbL self-assembly technology exhibits many advantages: (1) The operation process is a bottom-up assembly method, which can obtain controllable coating. (2) The utilization of LbL self-assembly technology to prepare multilayered coatings is simple, easy to operate, no need for special and complicated equipment and low cost. (3) Various components are suitable for LbL self-assembly technology to prepare multifunctional materials. (4) The preparation of multilayered coatings by LbL self-assembly technology is not limited by the size and shape of the substrate [31,32].

Over the past decades, the LbL self-assembly method to modify fabric has greatly developed due to the multifunctionalization of fabrics. Based on the research reports of the application of the LbL self-assembly method in fabric modification, we conducted a scientomeric review of the scientific literature relating to LbL self-assembly coatings on fabrics from 2005 to 2021.

## 2. Visualization of the Scientometrics Analysis of LbL Self-Assembly Coatings on Fabrics

In order to comprehensively analyze the work of the LbL self-assembly coatings on fabrics, related published works were investigated by bibliometric analysis and information visualization analysis. A bibliometric study is useful in obtaining an assessment of research or scientific production in a specific area over a period of time [33]. Information visualization analysis is a recent tool to detect and display bibliometric results. Some scholars have attempted to visualize bibliometric analysis results using network analysis or visual presentations using a combination of qualitative text analysis software [34]. CiteSpace is one of the most popular tools for knowledge mapping. This Java application is designed to visualize and analyze the trends and hotspots in a particular scientific research field, which was invented by Chen C.M. in early 2004 [35]. In this review, all bibliographic records of LbL self-assembly coating on fabric were published by the Web of Science (WOS) core collection database. Based on CiteSpace, the analysis results concluded four aspects according to different analytical parts, including growth trends analysis, keywords and keywords cluster analysis, country/region and institutions analysis.

### 2.1. Characteristics of Publication Outputs Analysis

The publication outputs can reveal the development situation in a certain field. The literature data were retrieved from WOS using the topics “layer-by-layer” and “coating” and “fabric”, and the scope was from 2005 to 2021. The original articles and reviews, published only in the English language, were selected, while ignoring proceedings and books. All results were manually reviewed to remove unrelated papers, and 238 articles and reviews were retrieved in plain text format with a full record, including cited references for scientometric analysis. The number of publication outputs for each year is shown in Figure 1. As presented in Figure 1, the evolution of the publication outputs can be divided into three stages. The first stage was from 2005 to 2009, when work on LbL self-assembly coatings on fabric emerged. During 2010–2014, the average number of publication outputs each year was approximately 10. The third stage was 2015–2021, and the research obviously increased. In these years, the publication numbers accounted for 76.5% of all results. From these results, it can be concluded that research on LbL self-assembly coatings on fabric has attracted increasing attention from scholars worldwide.

### 2.2. Keyword Co-Occurrence Analysis

Keywords are significant refinement, core content of the research article and keyword co-occurrence analysis for identifying critical research areas related to the work of LbL self-assembly coating on fabric [36]. Co-occurrence frequency and centrality are essential for visual knowledge mapping of keyword co-occurrence analysis by investigating the change over time in frequency and centrality [37]. Figure 2 shows the keyword co-occurrence network of the LbL self-assembly coating on fabric. It contains 282 nodes and 708 links. Each separate node signifies a keyword, while the keyword occurrence frequency shows the node, and the connection between the keywords symbolizes a link in the dataset [36]. The large size of the node represents the importance of the document, and the close connection symbolizes the high co-citation frequency.

The frequency of various keywords over time represents valuable information in the field of LbL self-assembly coating on fabric. The top 10 effective keywords in the LbL self-assembly coating on fabric ranked by frequency and centrality are listed in Table 1. The keywords “flammability”, “film” and “cotton fabric” have a higher frequency and centrality, indicating that researchers paid more attention to the related works. The keyword “cotton fabric” was highly influential on the scale of both centrality and frequency from 2011, which implies that many LbL self-assembly coatings were deposited on the surface of cotton fabric. Meanwhile, the keyword “flammability” showed high centrality and frequency since 2010, indicating that the flame retardant LbL self-assembly coating has been extensively investigated.

### 2.3. Keywords Cluster Analysis

As mentioned above, keywords are the core content of the article and the brief summary of the article. Herein, the retrieved correlative research was analyzed by the CiteSpace Ⅴ software. The keyword cluster analysis was performed with the CiteSpace clustering function from the keyword of the LbL self-assembly coating on fabric, and the 15 clusters are shown in Figure 3, which were labelled with the log-likelihood ratio (LLR) algorithm. This expresses how many times the data are more likely to be seen under one model than the other [38], and 15 knowledge clusters (based on keyword co-occurrence details) were extracted and shown in Figure 3. However, the auto-generated cluster could be misleading or too specific to clarify the meaning and scope. In order to further understand the major fields of research on LbL self-assembly coatings on fabric, the 15 clusters were further classified into three categories as follows:Matrix materials: This aspect mainly includes the specific fabric which contains the number of textiles, #6 polyester-cotton blend and #7 polyamide 66 textiles.Materials used in the coating: This section mainly contains the compositions of the coating, which include #4 bio-based coating, #8 polytetrafluoroethylene, #11 cellulose and #12 engineered nanoparticles.Functionalization of fabrics: This section mainly includes the properties of the modified fabrics. The properties relate to #1 flammability, #2 washability, #10 antibacterial surface, #13 underwater superoleophobicity, #14 thermal degradation properties and #15 electromagnetic shielding effectiveness.

### 2.4. Country/Territory/Institute Analysis

The global distribution of the LbL self-assembly coating on fabric was drawn based on CiteSpace in Figure 4. It contains 40 nodes and 46 links. Each separate node refers to a country/territory. The major research institutes are distributed in China, the United States and Italy. Moreover, the line in Figure 4 represents the cooperation of different counties/territories. China and the United States show more cooperation with other counties/territories. Table 2 lists the top 10 country/territory in LbL self-assembly coatings on fabric ranked by frequency. From Table 2, the publication outputs of China, the United States and Italy are 146, 46 and 26, respectively, and the centrality of the three countries is 0.39, 0.46 and 0.11, respectively. A higher centrality score implies that a country plays an important role in this research field [33]. Compared with China, the United States shows a higher importance with a lower frequency of publication outputs.

In order to further investigate the distribution of research institutes, the distribution was visualized with CiteSpace in Figure 5 and Table 3. It contains 195 nodes and 182 links. The top three most productive institutes were University Science and Technology of China, Politecnico di Torino and Texas A&M University, with 27, 25, and 19 publication outputs, respectively. From Figure 5, the related research has been concentrated in China in the past three years. Meanwhile, the total publication output of the top five most productive institutes is 104, accounting for 43.7% of all publications. This result illustrates that nearly half of the research results were carried out by these five institutes.

## 3. Functionalization of LbL Self-Assembly Coatings on Fabrics

Based on the keyword co-occurrence clusters analysis in Figure 3, the keyword clusters can be classified into three research themes. The third theme in Section 2.3 is the functionalization of fabrics, which includes #1 flammability, #2 washability, #10 antibacterial surface, #13 underwater superoleophobicity, #14 thermal degradation properties and #15 electromagnetic shielding effectiveness. Based on the classification, the functionalization of LbL self-assembly coatings on fabric was reviewed. This review highlights the composition of the coating and the functionalized properties achieved.

### 3.1. LbL Self-Assembly Coating to Reduce the Flammability of the Fabric

Based on the literature, the design of the flame retardant LbL self-assembly coatings starts from two perspectives, which are presented in Figure 6. The first is the coating acting as a barrier film based on inorganic nanoparticles [39]. The flame retardancy of fabrics is enhanced because the inorganic coatings act as thermal shielding barriers during combustion, which can effectively hinder the transmission of heat, oxygen, mass and volatile products from the underlying fabric to the atmosphere [40,41]. Many inorganic nanoparticles have been utilized in coatings, including montmorillonite [42,43,44,45,46], carbon nanotubes [45,47,48,49], silica [42,50,51,52,53] and titanium dioxide [54,55,56]. The second coating can form a stable carbon layer during combustion. Generally, phosphorus-containing polyelectrolytes and polyhydroxylated biomass polyelectrolytes are applied to fabricate flame-retardant coatings. During combustion, the stable carbon layer can also block the transmission of heat, oxygen, mass and volatile products [40]. The commonly used phosphorus-containing polyelectrolytes are ammonium polyphosphate [53,57,58,59,60,61], phytic acid [56,62,63,64,65,66,67,68], DNA [69], phosphatized cellulose [70], etc. The self-extinguishing effect can be achieved on the phosphorus-containing coating-modified fabric during the burning test.

Table 4 lists the different assembly types accompanied with the different coating increments, as well as different flame retardant properties. In 2009, Grunlan et al. at Texas A&M University reported branched polyethyleneimine (BPEI)/laponite clay platelets using the LbL self-assembly method on cotton fabric [71]. In the vertical burning test, the coated cotton fabric had more char residues than the uncoated cotton fabric. Thermogravimetric analysis (TGA) results revealed that the cotton fabric coated with 10 bilayers (BLs) of BPEI/laponite produced up to 6 wt% char residues at 500 °C. After this work, the LbL self-assembly technique was widely used to construct the flame-retardant coating on the surface of a polymer matrix. As mentioned in Section 2.4, the Politecn Torino ranked second among the institutes. In their work, α-zirconium phosphate nanoplatelets were alternatively combined with polydiallyldimethylammonium chloride (PDAC), polyhedral oligomeric silsesquioxane (POSS) or alumina coated silica (SiO_2_) nanoparticles to fabricate LbL self-assembly coatings on polyester fabrics [72]. In cone calorimetry tests, the three different deposited nanocoatings showed positive effects on reducing the flammability of polyester fabric. Then, the phosphorus-containing polyelectrolyte was introduced into the LbL self-assembly coating. The intumescent nanocoating was constructed using poly(sodium phosphate) (PSP) and poly(allylamine) (PAAm) on the surface of cotton fabric [73]. In the vertical burning test, fire could extinguish the fabric coated with 20 BLs of PSP/PAAm. Meanwhile, the peak heat release rate (PHRR) and total heat release of cotton resulted in a 43% and 51% reduction compared with the uncoated samples for microscale combustion calorimetry. An LbL self-assembled coating was prepared by the alternate adsorption of polyethylenimine (PEI) and ammonium polyphosphate (APP) on the surface of ramie fabric [74]. In the vertical burning test, the ramie fabric coated with 20 BL PEI/APP also achieved self-extinguishing. Polyallylamine (PAH) and polyphosphate (PSP) were alternatively deposited onto polyamide 6.6 fabrics [75]. The fabric coated with 40 BL PAH/PSP films resulted in a 36% reduction in PHRR in the cone test compared with the uncoated polyamide fabric. Hu et al. at the University of Science and Technology of China prepared hypophosphorous acid-modified chitosan (PCS)/BPEI multilayer coatings on the polyester-cotton fabric [76]. In a horizontal burning test, the afterglow phenomenon was eliminated, and self-extinguishing was observed for the fabric coated with the 20 BL coating. The same group also examined the effect of a hypophosphorous acid crosslinked PEI/oxide sodium alginate (OSA) multilayer coating on a polyethylene terephthalate (PET) fabric [77]. Before the deposition of the LbL self-assembled coating, the PET fabric was first grafted with acrylamide (AM) to ease the melt-dripping during burning. The combination of AM grafting and 15 BL crosslinked coating self-extinguished the fire without melt-dripping in the horizontal burning test. Moreover, the modified coating was durable for 12 laundering cycles. With the progress of research, studies have been focused on the multifunctionality of coatings. As a result, a multifunctional LbL self-assembled coating was developed on the surface of cotton fabric. The coated cotton fabric exhibited excellent flame retardancy, temperature sensing, fire-warning, piezoresistivity and joule heating performance [78]. This intelligent fireproof cotton fabric was prepared by coating MXene nanosheets with carboxymethyl chitosan (CCS). The MXene/CCS-modified cotton fabric showed accurate wide-range temperature sensing performance. The 4 BL MXene/CCS coating modified cotton fabric extinguished the fire in the vertical burning test. The coated fabric could repeatedly trigger the fire-warning system in less than 10 s when it burned. The fabric could also detect various human motions. This work suggests that fire resistant-coatings fabricated by the LbL self-assembly method exhibit great potential for reducing the fire hazard of fabrics, and the prepared coatings show promising alternatives to conventional flame retardants.

### 3.2. LbL Self-Assembly Coating to Endow the Fabric with an Antibacterial Surface

The LbL self-assembly method based on the regular alternating physical adsorption of anion and cation-charged polyelectrolytes is a low-cost and environmentally friendly process. This method can endow the substrate with a versatile performance by combining and controlling different molecules or particles [79]. Therefore, polyelectrolytes and nanoparticles with antibacterial effects were investigated to fabricate the LbL self-assembled coating. Table 5 shows the different assembly types accompanied with the different coating increments, as well as different antibacterial properties. In 2013, polycation chitosan (CH) and polyanion alginates (ALG) were alternatively deposited on the surface of cotton fabric using the LbL self-assembly technique [80]. The results showed that the cotton with five layers (CH/ALG/CH/ALG/CH) was more effective in inhibiting bacterial (*S. aureus* and *K. pneumanie*) growth. Demir et al. prepared pentasodium tripolyphosphate (TPP)/CH and poly(sodium 4-styrene sulfonate)/CH multilayers on cotton fabric [81]. The antibacterial effect increased with the number of layers on the surface of the fabric. The fabric coated with 15 BL TPP/CH showed 97% and 98% bacterial reductions against *S. aureus* and *K. pneumanie*, respectively. A novel silicon-and nitrogen-containing compound, poly [3-(5,5-cyanuricacidpropyl)-siloxane-co-trimethyl ammoniumpropyl siloxane chloride] (PCQS), was synthesized. Then, PCQS, PEI and phytic acid (PA) were deposited on cotton fabrics with the LbL self-assembly method [82]. The cotton coated with 30 BL PEI/(PCQS/PA) showed excellent flame retardancy and antibacterial activity. Cotton-PEI/(PCQS/PA)30-Cl exhibited effective antimicrobial activity against *E. coli* O157:H7 and *S. aureus* with 100% bacterial reduction within 1 min of contact time. Poly(styrenesulfonate) (PSS) and synthesized silver-loaded chitosan (CS-Ag) nanoparticles were alternatively deposited on the surface of the cotton fabric [83]. The cotton coated with 15 BL PSS/CS-Ag exhibited 100% bactericidal activity against *S. aureus* and *E. coli*. Saini et al. constructed CH and green tea extract (GTE) multilayered films on linen fabric [84]. The obtained fabric showed effective antioxidant properties, antibacterial activity and UV protection. The 10 BL CH/GTE coating on linen fabric resulted in 97% antibacterial activity against *S. aureus.* Chitosan-cinnamaldehyde crosslinked nanoparticles (CSN) were synthesized and combined with TPP to fabricate an LbL self-assembled coating on PET nonwoven fabric [85]. The nanolevel coating on the fabric with 10.5 BLs (0.5 BL refers to CSN) exhibited remarkable antibacterial activity against *S. aureus* and *E. coli*. In addition, the obtained fabric could adsorb metal ions, such as Pb^2+^, Cu^2+^, Fe^3+^, Ni^2+^, Zr^4+^ and Cd^2+^, from water. All these coatings belong to polyelectrolyte multilayers. The positive charges in the coatings can disrupt bacterial membranes and inhibit bacterial adhesion and proliferation. The LbL self-assembly method can be used to create the antibacterial surface on any charged fiber, requiring simple and inexpensive procedures.

### 3.3. LbL Self-Assembly Coating on the Fabric to Shield Ultraviolet (UV) Light

UV-A (315–400 nm) and UV-B (290–315 nm) can reach the Earth’s surface and cause serious health problems, such as skin cancer, sunburn and photo-aging. Thus, the UV transmission of textiles has attracted much attention. Protective clothing should have UV-reflecting and/or absorbing properties as high as possible to prevent UV rays from reaching the skin and threatening the human body [86]. In previous studies, UV-absorbing chemicals were utilized to finish the fabrics [87,88,89,90,91]. Among the finishing methods, the LbL self-assembly technique was utilized to fabricate a UV shielding multilayer film on the surface of the fabric, as presented in Figure 7. The components in the LbL self-assembly coating can reflect or absorb UV light to endow fabrics with UV shielding properties, which is presented in Table 6. Three fluorescent brightening agents (FBAs) and poly(diallyldimethylammonium chloride) (PDDA) were alternatively constructed on cotton fabrics through the LbL deposition technique [92]. Among the three FBA agents, 10 BLs of Uvitex NFW/PDDA-coated cotton exhibited the highest ultraviolet protection factor (UPF) (>70). Additionally, the multilayers showed excellent durability in the washing test. ZnO nanoparticles have also been utilized to fabricate LbL self-assembled coatings on cotton woven fabric [93]. The UPF factor of untreated cotton fabric was 4.16, while the UPF factor of cotton treated with the 16 nano-ZnO multilayer coating was 12.3. Moreover, the ZnO coating on the cotton showed excellent antimicrobial activity against *S. aureus* bacteria. Zhao et al. prepared a hybrid layered double hydroxide (LDH) nanoplatelet modified with 2-hydroxy4-methoxybenzophenone-5-sulfonic acid (HMBS) and 3-aminopropyltriethoxy silane [94]. Then the modified LDH was combined with poly(acrylic acid) (PAA) to form multilayers on the surface of cotton fabric. Compared with untreated fabric, the UPF of cotton with the 5 BL PAA/modified PAA coating increased from 3.7 to 15.5 because the organic UV absorbers (HMBS) were intercalated into the LDH layers. Cotton fabrics with UV shielding properties were prepared by depositing graphene oxide (GO) and CH on cotton fabric via LbL self-assembly. The cotton fabric with 10 BLs GO/CH presented a UPF value of 452 compared to that of the untreated cotton (UPF = 9.37). The remarkable enhancement of UV protection may be due to the deeper color caused by GO. Moreover, the coated cotton fabric exhibited excellent washing durability after 10 rounds of water laundering. In Xiong’s work, PEI-H was synthesized by grafting PEI with 2,4-dihydroxybenzophenone. Then PEI-H and silica were sequentially deposited onto the cotton fabric. The results showed that the obtained cotton fabric exhibited superior UV resistance with a UPF value of 876.13, which could be attributed to the synergistic effect between the organic UV absorber (2,4-dihydroxybenzophenone) and the inorganic UV shielding agent (SiO_2_). The cotton fabric was treated using the LbL self-assembly method using a solution of CH and sodium lignin sulphonate (SLS) with boric acid (BA) [95]. The UPF values increased with more bilayers. The cotton coated with 3 BLs presented a 77.52 UPF value. Additionally, the CH/SLS-BA-treated cotton showed wrinkle-free, antibacterial, flame-retardant and antioxidant properties.

### 3.4. LbL Self-Assembly Coating to Prepare Hydrophobic Fabric

The LbL self-assembly method can create a hydrophobic surface due to the advantage of being able to tailor the surface morphology of the nanostructures by controlling the assembly cycles [97]. Most of the components used in the LbL self-assembled coating are hydrophilic. As shown in Figure 8, the hydrophobic layers were usually deposited after the construction of the LbL self-assembled coating [53,98,99,100,101,102]. Thus, the coating fabricated by the LbL self-assembly method aims to create rough surfaces on fabric. Table 7 shows the different assembly types accompanied with the different coating increments, as well as different hydrophobic properties. The superhydrophobic cotton was fabricated by using cationic poly (dimethyldiallylammonium chloride) (Poly-DMDAAC) and silica particles with subsequent modification of (heptadecafluoro-1,1,2,2-tetradecyl) trimethoxysilane [20]. The coated fabric showed excellent chemical stability and nonwettability with a water contact angle (WCA) of 155°. The PET fabrics were functionalized by introducing carbon nanotubes (CNTs) on the surface of the PET fiber by LbL self-assembly using poly(dimethyl diallyl ammonium chloride) (PDDA) as a polyelectrolyte, followed by post-treatment with poly(dimethylsiloxane) (PDMS) [103]. The PET fabric coated with 10 BL PDDA/CNT multilayers and further treated with PDMS presented a superhydrophobic surface with a WCA of 166.9°. The treated fabrics showed resistance to acid/alkaline etching, UV irradiation, long-term laundering and mechanical abrasion. Additionally, the multilayer coating exhibited UV blocking properties and electrical conductivity. Xiong et al. prepared an ultraviolet UV absorber (PEI-H) [16], in which PEI-H was combined with SiO_2_ and hexadecyltrimethoxysilane (HDTMS) to fabricate multilayered films on the surface of cotton fabric. The obtained cotton showed a superhydrophobic surface with a WCA of 154°, and the hydrophobic surface was durable after 20 washing times or 300 abrasion times. As mentioned in Section 3.3, the PEI-H/SiO_2_-based coating exhibited UV shielding properties. Mixed polyelectrolytes, including branched poly(ethylenimine) (BPEI), phytic acid (PA) and ammonium polyphosphate (APP), were deposited on the PET fabric, thereby resulting in an excellent flame retardant. After the flame-retardant fabric was further treated with a layer of PDMS-grafted-TiO_2_@PDMS coating, the fabric exhibited superhydrophobicity and self-cleaning properties with a WCA of 162° under air conditions. Moreover, the fabric exhibited photocatalysis capability and photocatalytically stable superhydrophobicity. A superhydrophobic photothermal conversion cotton fabric was prepared by LbL self-assembly of CNTs on the surface of fibers, followed by post-treatment with PDMS [10]. The as-obtained cotton fabric presented superhydrophobicity with a WCA of 165° and a rolling angle of 0.6°. Moreover, the treated cotton could be rapidly heated to 89.8 °C under one sun (1 kW/m^2^) irradiation.

### 3.5. Electromagnetic Interference Shielding LbL Self-Assembly Coating on the Fabric

Electromagnetic radiation has become the fourth most serious source of pollution after noise, water and air pollution, due to the advancement of electrification and the application of electromagnetic energy [104]. Fabrics are intrinsically electrical insulating materials and nonelectromagnetic interference (EMI) shielding materials. Thus, it is important to develop textile materials that can protect the human body or sensitive electronic equipment from electromagnetic waves. For EMI shielding of LbL self-assembled coatings, many nanoparticles and polyelectrolytes have been utilized, such as NiFe_2_O_4_ nanoparticles, graphene, CNTs, MXene, polypyrrole and polyaniline [105,106,107,108,109]. Table 8 presents the different assembly types accompanied with the different coating increments, as well as different electromagnetic interference shielding properties. A CH/graphene mixed solution as a polycation and poly(sodium 4-styrenesulfonate) (PSS) as a polyanion were deposited on cotton fabric by LbL self-assembled coating [109]. The results showed that the 10 BL coating-treated fabric had an electrical conductivity of 1.67 × 10^3^ S/m. The cotton fabric exhibited an EMI shielding ability with a maximum shielding effectiveness (SE) of 30.04 dB. High-loading CNT/poly(allylamine hydrochloride) (PAH) multilayers were constructed on the cellulose fabric [108]. The coated fabrics exhibited an EMI-SE of 11.9 dB and were still soft, flexible and air-permeable. An eco-friendly flame retardant and an EMI shielding coating were fabricated by depositing PEI/PA multilayers and silver nanowire (AgNWs) conductive layers on cotton fabric [110]. The cotton with 24.2 wt% of PEI/PA-based coating and 7.5 wt% of AgNWs performed a self-extinguishing effect and a 58.59% reduction in the PHRR value. Additionally, the flame-retardant cotton showed a high electrical conductivity of 2416.46 S/m and an EMI-SE of 32.98 dB. This coating was durable during the bending, washing and sandpaper abrasion tests. Plain fabrics with different fabric densities were weaved and used as the substrate to be LbL self-assembled by graphite oxide (GO) and polypyrrole (PPy) [111]. Compared with pure cotton fabric, the EMI SE of the coated fabric increased by approximately 71%. Moreover, the EMI-SE was always the maximum at a fabric density of 100 × 100 picks/100 cm. Similar work was also reported on a GO/PPy multilayered coating on cotton fabric [112], which illustrated how the coating structure affected the EMI shielding performance. The results showed that more GO/PPy interfaces in the coating resulted in stronger EMI shielding enhancement because the conductive network was built. Yin et al. fabricated a 1D polyaniline (PANI) and 2D MXene nanosheets-based multilayered coating on carbon fiber fabric followed by a PDMS layer [106]. The as-prepared fabric exhibited a high electrical conductivity of approximately 325 S/m and an EMI-SE of approximately 35.3 dB with 0.376 mm. Additionally, the hydrophobic surface of the carbon fiber fabric was created by depositing PDMS. The WCA of the treated fabric could reach 135.2°.

### 3.6. Wash-Durability of the LbL Self-Assembly Coating on the Fabric

Most of the components used in the LbL self-assembly coating on the fabric were water-soluble or hydrophilic, resulting in poor wash durability for the coatings on the fabrics [113]. Thus, durable coatings on fabric are urgently required and cannot be ignored. To improve water durability, two methods are usually applied. The first is to establish a connection between layers. Carosio et al. prepared a UV-curable aliphatic acrylic polyurethane latex doped with APP and chitosan multilayered coating on cotton fabric [114,115]. Then, the treated cotton was exposed to UV radiation, resulting in a coating in which APP was in intimate contact with chitosan within a UV-cured network. The results showed that the modified cotton fabric self-extinguishment in horizontal flame spread tests was achieved after washing in water at 65 °C for 1 h. The second method is the introduction of a hydrophobic surface onto the fabric to reduce the need for fabric laundering and resolve the leaching of the functionalized components during laundering [16]. A trilayer of bPEI, APP and fluorinated-decyl polyhedral oligomeric silsesquioxane (F-POSS) is fabricated on cotton [113]. The treated cotton fabric could extinguish the fire when directly exposed to flame and could generate an intumescent char layer. The F-POSS embedded in cotton fabric and the APP/bPEI coating produced a superhydrophobic surface with a self-healing function.

## 4. Conclusions

This bibliometrics-based review summarizes recent research progress on LbL self-assembled coatings for functionalized fabrics, which was extracted from the WOS from 2005 to 2021. The publication numbers on LbL self-assembly coatings on fabric during 2015–2021 accounted for 76.5% of all publications, suggesting that research has been rapidly developed in recent years. From the visualization analysis, the keywords “flammability”, “film” and “cotton fabric” have higher frequency and centrality. In the keyword co-occurrence clusters analysis, the functionalization of fabric contains #1 flammability, #2 washability, #10 antibacterial surface, #13 underwater superoleophobicity, #14 thermal degradation properties and #15 electromagnetic shielding effectiveness.

Based on the classification, the functionalization of LbL self-assembly coatings on fabric was also reviewed. From the review results, it can be concluded that the recent research developments of LbL self-assembly coatings on fabric have been mainly focused on the following aspects: (1) According to the required functionalization, a suitable composition of the coating can be chosen, including organic and inorganic components. (2) The functionalization of coatings on fabric develops from singularity to diversification. (3) Wash-durability and wear resistance are considered during the preparation of the LbL self-assembled coating.

Meanwhile, based on the review on the LbL self-assembly coating on fabrics, it can be summarized that the following issues should be investigated: (1) Due to the weak interaction between layers in the LbL self-assembly coating, the durability of the coating is poor. In order to improve the wash-durability and wear resistance, more methods need to be utilized to enhance the bonds between the layers. (2) For the finishing of fabrics, comfort cannot be ignored. Only a limited number of papers focused on the breathability and softness of the modified fabrics. Thus, future investigations should take into account fabric comfort for further application. (3) Different components can endow fabric with abundant properties. Moreover, LbL self-assembly is a green and low-cost method. It is important to find environmentally friendly and inexpensive components to create more functionality for the fabrics.

## Figures and Tables

**Figure 1 molecules-27-06767-f001:**
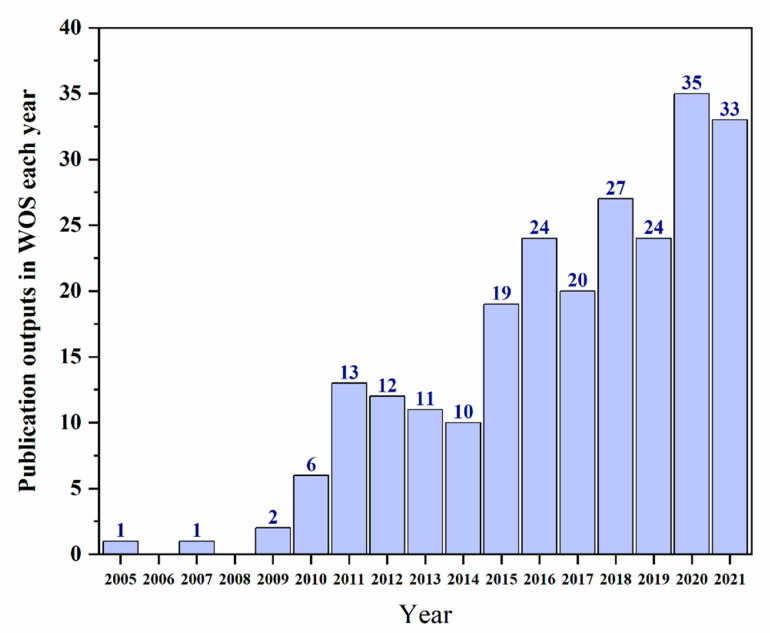
Change in the number of research studies on LbL self-assembly coatings on fabric change with time.

**Figure 2 molecules-27-06767-f002:**
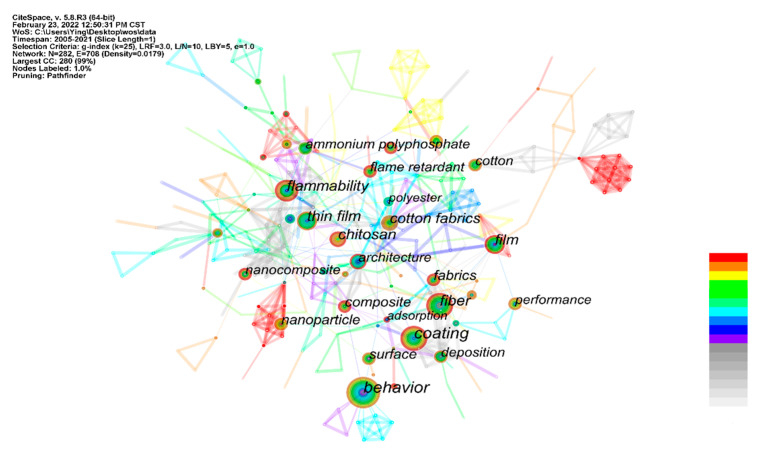
Knowledge mapping of the keyword co-occurrence network of the LbL self-assembly coating on fabric.

**Figure 3 molecules-27-06767-f003:**
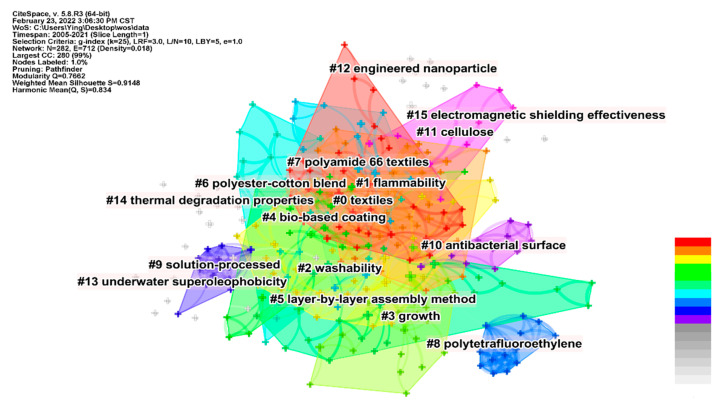
Co-occurring keywords-based knowledge clusters.

**Figure 4 molecules-27-06767-f004:**
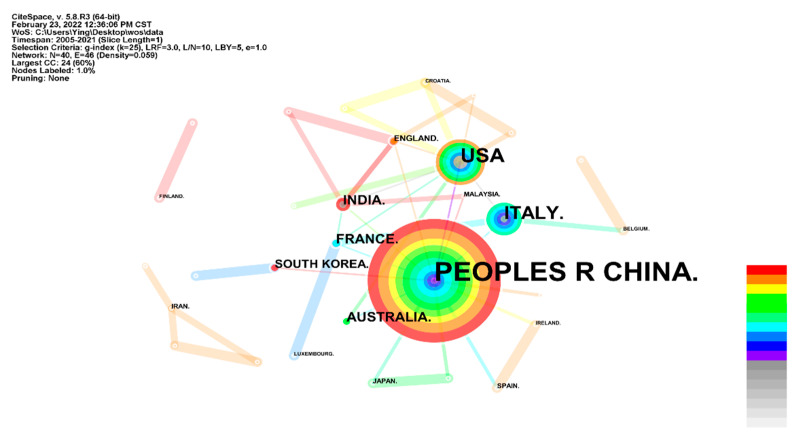
Country co-authorship network in research on LbL self-assembly coating on fabric.

**Figure 5 molecules-27-06767-f005:**
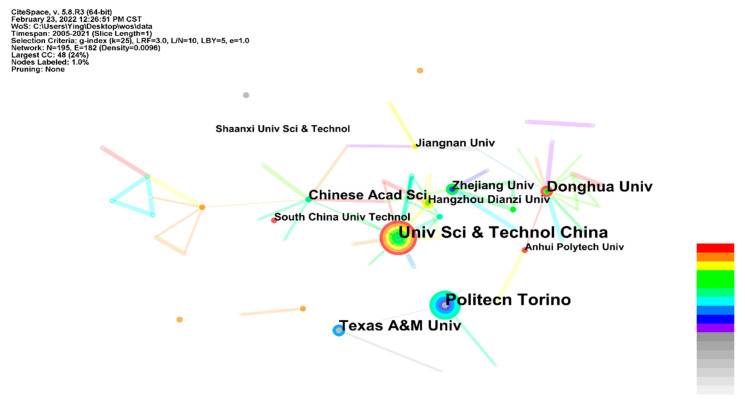
Institute co-authorship network in LbL self-assembly coating on fabric research.

**Figure 6 molecules-27-06767-f006:**
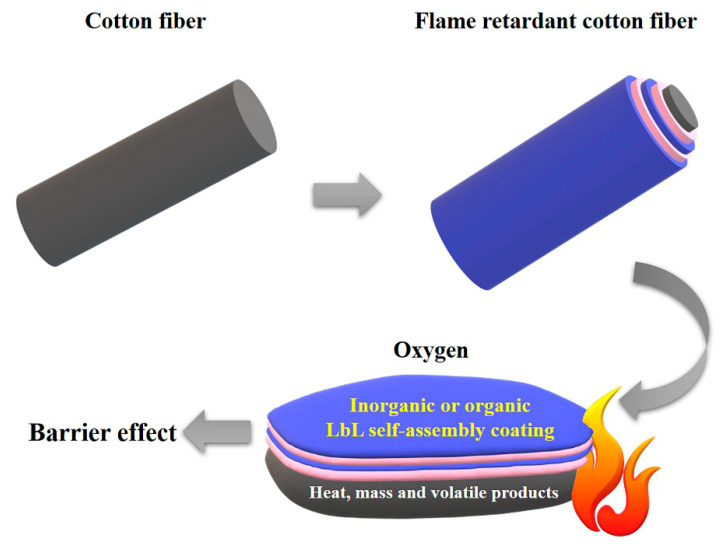
Schematic of flame-retardant mechanism of LbL self-assembly coating.

**Figure 7 molecules-27-06767-f007:**
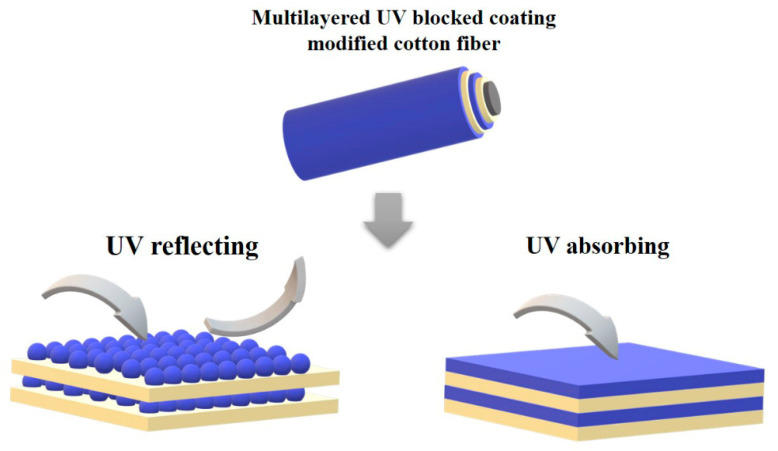
Schematic of UV shielding of LbL self-assembly coating.

**Figure 8 molecules-27-06767-f008:**
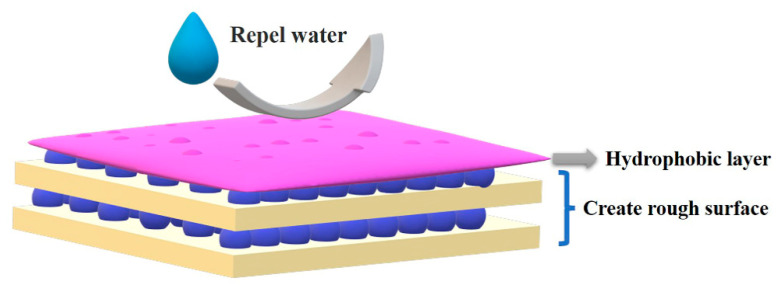
Schematic of LbL self-assembly method to create the hydrophobic surface.

**Table 1 molecules-27-06767-t001:** Top 10 effective keywords in LbL self-assembly coating on fabric ranked by frequency and centrality.

Frequency	Centrality	Year	Keyword
50	0.09	2009	coating
48	0.11	2011	behavior
38	0.16	2010	flammability
38	0.01	2012	chitosan
38	0.07	2010	fiber
38	0.15	2011	thin film
32	0.19	2010	film
31	0.24	2011	cotton fabrics
28	0.09	2010	fabrics
27	0.02	2010	nanoparticle

**Table 2 molecules-27-06767-t002:** Top 10 countries/territories in LbL self-assembly coating on fabric ranked by frequency.

Frequency	Centrality	First Occurrence	Country
146	0.39	2010	PEOPLES R CHINA.
46	0.46	2009	USA
26	0.11	2011	ITALY.
12	0.29	2011	INDIA.
11	0	2010	AUSTRALIA.
10	0.21	2014	FRANCE.
7	0.06	2011	SOUTH KOREA.
5	0	2010	TURKEY.
4	0	2012	THAILAND.
4	0.36	2020	ENGLAND.

**Table 3 molecules-27-06767-t003:** Top 10 institutes in research on LbL self-assembly coating on fabric ranked by frequency.

Frequency	Centrality	First Occurrence	Institution
27	0.05	2014	Univ. Sci & Technol. China
25	0.01	2011	Politecn. Torino
19	0.01	2009	Texas A&M Univ.
19	0.05	2012	Donghua Univ.
14	0.09	2015	Chinese Acad. Sci.
10	0.02	2013	Zhejiang Univ.
8	0.01	2018	Hangzhou Dianzi Univ.
8	0.05	2010	Jiangnan Univ.
7	0	2016	South China Univ. Technol.
6	0	2019	Anhui Polytech Univ.

**Table 4 molecules-27-06767-t004:** Flame retardant effect of the LbL self-assembly coating on the fabric.

Fabric	Year	Composition	Layers	Best Burning Test Result		Reference
				Horizontal Burning Test	Vertical Burning Test	
Cotton	2009	BPEI/Laponite	10 BL	-	More char residue	[71]
Polyester	2011	α-ZrP/PDAC α-ZrP/POSS α-ZrP/SiO_2_	10 BL	-	-	[72]
Cotton	2011	PSP/PAAm	20 BL	-	Self- extinguish	[73]
Ramie	2013	PEI/APP	20 BL	-	Self- extinguish	[74]
Polyamide 6.6	2014	PAH/PSP	40 BL	-	-	[75]
Polyester-cotton blends	2017	PCS/BPEI	20 BL	Self- extinguish	-	[76]
Polyethylene terephthalate	2019	PEI/OSA/HA	15 BL	Self- extinguish	-	[77]
Cotton	2021	MXene/CCS	4 BL	Self- extinguish	-	[78]

**Table 5 molecules-27-06767-t005:** Antibacterial effect of LbL self-assembly coating on the fabric.

Fabric	Year	Composition	Layers	Bacterial	Reference
Cotton	2013	CH/ALG	CH/ALG/CH/ALG/CH	*S. aureus* and *K. pneumanie*	[80]
Woven cotton	2017	TPP/CHT and PSS/CHT	15 BL	*S. aureus* and *K. pneumanie*	[81]
Cotton	2019	PCQS/PA	30 BL	*E. coli* O157:H7 and *S. aureus*	[82]
Cotton	2020	PSS/CS-Ag	15 BL	*S. aureus* and *E. coli*	[83]
Linen	2020	CH/GTE	10 BL	*S. aureus*	[84]
PET nonwoven	2021	CSN/TPP	10.5	*S. aureus* and *E. coli*	[85]

**Table 6 molecules-27-06767-t006:** UV resistance of the LbL self-assembly coating on the fabric.

Fabric	Year	Composition	Layers	UPF	Reference
Cotton	2010	FBAs/PDDA	10 BL	>70	[92]
Cotton	2010	ZnO/ZnO	16 BL	12.3	[93]
Cotton	2013	PAA/modified LDH	5 BL	15.5	[94]
Cotton	2016	GO/CH	10 BL	452	[96]
Cotton	2019	PEI-H/SiO_2_/HDTMS	5 BL	876.13	[16]
Cotton	2020	CH/SLS-BA	3 BL	77.52	[95]

**Table 7 molecules-27-06767-t007:** LbL self-assembly coating to create a hydrophobic surface on the fabric.

Fabric	Year	Composition of LbL Self-Assembled Coating	Hydrophobic Layer	Layers	WCA	Reference
Cotton	2012	Poly-DMDAAC/SiO_2_	(heptadecafluoro-1,1,2,2-tetradecyl) trimethoxysilane	2 BL	155°	[20]
PET	2017	PDDA/CNT	PDMS	10 BL	166.9°	[103]
Cotton	2019	PEI-H/SiO_2_	HDTMS	3 BL	154°	[16]
PET	2020	BPEI/PA/APP	PDMS-grafted-TiO_2_@PDMS	2 BL	162°	[55]
Cotton	2021	CH/CNT	PDMS	10 BL	165°	[10]

**Table 8 molecules-27-06767-t008:** EMI shielding LbL self-assembly coating on fabric.

Fabric	Year	Composition	Layers	EMI SE	Electrical Conductivity	Reference
Cotton	2017	CH+graphene/PSS	10 BL	30.04 dB	1670 S/m	[109]
Cellulose	2018	CNT/PAH	30 BL	11.9 dB	-	[108]
Cotton	2019	PEI/PA/AgNWs	8 BL PEI/PA + 4 layers AgNWs	32.98 dB	2416.46 S/m	[110]
Cotton	2021	GO/PPy	5 BL	19.2 dB	-	[111]
Cotton	2021	GO/PPy	4 BL	39.1 dB	-	[112]
Carbon fiber	2021	PANI/MXene/PDMS	50 BL	35.3 dB	325 S/m	[106]

## Data Availability

Data sharing not applicable.

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
