# Peer review of "Layer-by-Layer Self-Assembly Coating for Multi-Functionalized Fabrics: A Scientometric Analysis in CiteSpace (2005–2021)"

_molecules, 2022, doi:10.3390/molecules27196767_

Round 1

Reviewer 1 Report

English accuracy should be improved, so as to enhance the readability of the manuscript. Grammatical errors have to be corrected.

Author Response

Dear reviewer,

Thank you for your suggestion and comments on our manuscript. We have revised the manuscript thoroughly according to the opinions of reviewers. Now the manuscript, Layer-by-Layer Self-assembly Coating for Multi-functionalized Fabrics: A Scientometric Analysis in CiteSpace (2005–2021), has been uploaded. The reviewer’s comments are processed as follows.

If you have any question about the manuscript, please do not hesitate to contact us by e-mail.

Yours sincerely,

Ying Pan

------------------------------------

Comments and Suggestions for Authors:

English accuracy should be improved, so as to enhance the readability of the manuscript. Grammatical errors have to be corrected.

Reply: Thanks a lot for your review work! The manuscript has been carefully modified.

Reviewer 2 Report

The review article (Manuscript Number molecules-1907640) describes the crucial advances concerning common methodologies for the (among others) layer-by-layer self-assembly coating for multi-functionalized fabrics for flame retardant fabric, antibacterial fabric, ultraviolet resistant fabric, hydrophobic fabric, and electromagnetic shielding fabric. As the authors stated, the analysis of the literature provides some points on which the investigations should be mainly focused, in particular, the identification of thresholds in the acceleration factors, but they don’t mention other polymer preparation approaches.
On the whole, the manuscript is fairly well-written and logically arranged. The overall originality of the review concept used here is medium-high. Nevertheless, I would recommend the publication of this paper in Molecules on the condition a major revision of the manuscript will be carried out and the following points will be taken into consideration.

Detailed comments:
1. After reading this present manuscript, I would like to recommend the authors write a real introduction to the potential materials that can be prepared on the basis of this strategy. (short introduction about, origin, synthesis, physicochemical properties, polymer family, biodegradability...).
2. More detailed advantages of the present field must be mentioned in the Introduction.
3. Furthermore, the Introduction should be worked out - so as to show the full state of knowledge on different polymerization methods, as the opposite solutions. Extension to look at these issues and also provide other techniques should also be provided. I recommend the following literature being cited: 10.1021/acs.macromol.7b01730 (Macromolecules 2017, 50, 8417-8425), 10.1002/macp.201900285 (Macromolecular Chemistry and Physics 2019, 220, 1900285), 10.3390/ma13071717 (Materials 2020, 13, 1717), 10.1002/app.49275 (Journal of Applied Polymer Science 2020, 137, 49275) and 10.1002/pat.4921 (Polymers for Advanced Technologies 2020, 31, 1972-1979). One of the most important is the ATRP strategy when you think about future “biological properties” including self-cleaning surfaces so huge catalyst loading is not appropriate to choose. But, on the other side, it should be noticed that if authors consider showing the application of one of the low-ppm ATRP methods or just metal-free ATRP - this study should be more useful and valuable for the readers of another journal. It creates a real possibility of receiving polymers with precisely designed molecular weight and molecular weight distribution.
4. The abstract needs to be well written with future prospects of the work and describe in short the long-term behavior of polymers exposed to different environments.
5. The conclusion reflects an overall summary of the field with further extension and includes future perspectives - I would suggest clarifying this section.
6. The article appears to be a collection of data from research papers, however, the author's self-opinion is of importance while drafting a manuscript of this type. Therefore, the presented results are semi-informative, and the discussion is not clear.
7. The style and grammar leave much to be desired in many places, some parts of the text are difficult to understand.

After completing the above-mentioned corrections this work will be more readable. Therefore, it will be useful for the readers of the Molecules.

Author Response

Dear reviewer,

Thank you for your valuable suggestion and comments on our manuscript. We have revised the manuscript thoroughly according to the opinions of reviewers. Now the manuscript, Layer-by-Layer Self-assembly Coating for Multi-functionalized Fabrics: A Scientometric Analysis in CiteSpace (2005–2021), has been uploaded. The editor’s comments are processed point by point as follows.

If you have any question about the manuscript, please do not hesitate to contact us by e-mail.

Yours sincerely,

Ying Pan

------------------------------------

Comments and Suggestions for Authors:

The review article (Manuscript Number molecules-1907640) describes the crucial advances concerning common methodologies for the (among others) layer-by-layer self-assembly coating for multi-functionalized fabrics for flame retardant fabric, antibacterial fabric, ultraviolet resistant fabric, hydrophobic fabric, and electromagnetic shielding fabric. As the authors stated, the analysis of the literature provides some points on which the investigations should be mainly focused, in particular, the identification of thresholds in the acceleration factors, but they don’t mention other polymer preparation approaches.

On the whole, the manuscript is fairly well-written and logically arranged. The overall originality of the review concept used here is medium-high. Nevertheless, I would recommend the publication of this paper in Molecules on the condition a major revision of the manuscript will be carried out and the following points will be taken into consideration.

Detailed comments:

  1. After reading this present manuscript, I would like to recommend the authors write a real introduction to the potential materials that can be prepared on the basis of this strategy. (short introduction about, origin, synthesis, physicochemical properties, polymer family, biodegradability...).

Reply: Thanks for your careful review work! Now the potential materials that can be prepared on the basis of this strategy have been added in the introduction.

  1. More detailed advantages of the present field must be mentioned in the Introduction.

Reply: Thanks for your careful review work! The more detailed advantages of the present field have been added in the introduction. Compared with traditional finishing methods, LbL self-assembly technology exhibits many advantages: (1) The operation process is a bottom-up assembly method, which can obtain controllable coating. (2) The utilization of LbL self-assembly technology to prepare multilayered coatings is simple, easy to operate, no need for special and complicated equipment, and low cost. (3) Various components are suitable for LbL self-assembly technology to prepare multi-functional materials. (4) The preparation of multilayered coating by LbL self-assembly technology is not limited by the size and shape of the substrate.

  1. Furthermore, the Introduction should be worked out - so as to show the full state of knowledge on different polymerization methods, as the opposite solutions. Extension to look at these issues and also provide other techniques should also be provided. I recommend the following literature being cited: 10.1021/acs.macromol.7b01730 (Macromolecules 2017, 50, 8417-8425), 10.1002/macp.201900285 (Macromolecular Chemistry and Physics 2019, 220, 1900285), 10.3390/ma13071717 (Materials 2020, 13, 1717), 10.1002/app.49275 (Journal of Applied Polymer Science 2020, 137, 49275) and 10.1002/pat.4921 (Polymers for Advanced Technologies 2020, 31, 1972-1979). One of the most important is the ATRP strategy when you think about future “biological properties” including self-cleaning surfaces so huge catalyst loading is not appropriate to choose. But, on the other side, it should be noticed that if authors consider showing the application of one of the low-ppm ATRP methods or just metal-free ATRP - this study should be more useful and valuable for the readers of another journal. It creates a real possibility of receiving polymers with precisely designed molecular weight and molecular weight distribution.

Reply: Thanks for your careful review work! The literatures mentioned have been cited in the introduction. In the future work, ATRP strategy can be applied to create the multifunctional coating on the surface of fabric.

  1. The abstract needs to be well written with future prospects of the work and describe in short the long-term behavior of polymers exposed to different environments.

Reply: Thanks for your careful review work! Now the future prospects of the work and long-term behavior of polymers exposed to different environments have been added in the abstract.

  1. The conclusion reflects an overall summary of the field with further extension and includes future perspectives - I would suggest clarifying this section.

Reply: Thanks for your careful review work! The further extension has been supplemented in the conclusion. Meanwhile, based on the review about the LbL self-assembly coating on fabrics, it can be summarized that the following issues should be investigated: 1) Due to the inter-action between layers in the LbL self-assembly coating is weak, the durability of the coating is poor. In order to improve the wash-durability and wear resistance, more methods need to be utilized to enhance the bonds between the layers. 2) For the finishing of fabrics, comfort can’t be ignored. Only a limited number of papers focused on the breathability and softness of the modified fabrics. Thus, more future investigations should be taken into account on fabric comfort for further application. 3) Different components can endow fabric with abundant properties. Moreover, LbL self-assembly is a green and low-cost method. It is meaningful to find environmentally friendly and inexpensive components to create more functionality on the fabrics.

  1. The article appears to be a collection of data from research papers, however, the author's self-opinion is of importance while drafting a manuscript of this type. Therefore, the presented results are semi-informative, and the discussion is not clear.

Reply: Thanks for your careful review work! Now we added more our opinion in the manuscript.

  1. The style and grammar leave much to be desired in many places, some parts of the text are difficult to understand.

Reply: Thanks a lot for your review work! The manuscript has been carefully modified.

After completing the above-mentioned corrections this work will be more readable. Therefore, it will be useful for the readers of the Molecules.

Reviewer 3 Report

Interteresting review Check at Page 4,line 122, the Number relates to polytetrafluoroborate Is omissis

Author Response

Dear reviewer,

Thank you for your valuable suggestion and comments on our manuscript. We have revised the manuscript thoroughly according to the opinions of reviewers. Now the manuscript, Layer-by-Layer Self-assembly Coating for Multi-functionalized Fabrics: A Scientometric Analysis in CiteSpace (2005–2021), has been uploaded. The editor’s comments are processed point by point as follows.

If you have any question about the manuscript, please do not hesitate to contact us by e-mail.

Yours sincerely,

Ying Pan

------------------------------------

Comments and Suggestions for Authors:

Interesting review Check at Page 4, line 122, the Number relates to polytetrafluoroborate is missed.

Reply: Thanks a lot for your review work! The missing part has been added at Page 4, Line 122.

Round 2

Reviewer 1 Report

The revised manuscript can be accepted.

Reviewer 2 Report

I would like to support this revised paper (Manuscript Number molecules-1907640) for publication in the Molecules. All suggested changes were made (or discussed/clarified) by the authors. The results are informative, and the discussion is clear. To summarize, I think that this paper can be published as-is.